# Financing innovation and enterprises' efficiency of technological innovation in the internet industry: Evidence from China

**Zhefan Piao, Yueqin Lin**⬥*

School of Finance, Zhejiang University of Finance & Economics, Hangzhou, China

* adbhcew@163.com

**Data Availability Statement:** All relevant data are within the paper and its Supporting Information files.

**Funding:** Initials of the authors who received each award: PZF; Specific grant numbers: 300000;

## Abstract

This study empirically examined the impact of financing innovation on technological innovation efficiency of select internet companies, that were affiliated with China between 2008 and 2017. Analysis was based on their patent and annual report data and used multiple input-output SFA model, system GMM, and panel fixed-effect model. The results are as follows. (1) There is significant variation in overall technological innovation efficiency of listed companies in the internet industry, and there is a downward trend. The technological innovation efficiency of business that use financing innovation methods is higher than those that do not. (2) The number of patents and intangible capital investment of internet businesses increase obviously every year, but there is no corresponding increase in the efficiency of technological innovation, and little intangible capital investment of non-financing innovation businesses. Thus, determining how to effectively improve the overall quality of patents and the efficiency of intangible capital investment is essential to improve the efficiency of technological innovation for Chinese internet businesses. (3) There is a term mismatch in the investment and financing of internet businesses in China. The financing structure between the financing innovation and non-financing innovation businesses has different impacts on the efficiency of technological innovation. And nowadays, more financing channels are short-term debt financing channels which invest in projects to improve the efficiency of technological innovation due to the pressure of debt repayment and the need to protect shareholders' interests. (4) In the panel regression, the coefficients of *Icd* and *Roa* are significantly negative, suggesting that the investment efficiency of internet businesses needs to be improved.

## Introduction

With the continuous upgrading and development of the internet industry and its deepening integration with economy and society, the internet has become the main driving force of global economic growth. China's internet industry has developed rapidly. According to the China Industrial Research Report Network, in 2016, China's digital economy, including the internet, reached 22.6 trillion yuan, making it second in the world after the United States, with a

Funder: The Key Program of Humanities and Social Science of Colleges in Zhejiang Province (No. 2016ZB003); URL of funder website: http://jyt.zj. gov.cn/ The funders had no role in study design, data collection and analysis, decision to publish, or preparation of the manuscript.

**Competing interests:** The authors have declared that no competing interests exist.

nominal growth of more than 18.9% and a contribution of 69.9% to GDP growth [1]. China's top 100 internet companies, headed by Alibaba, Tencent, Baidu, Jingdong, NetEase, Sina, and Sohu, increased their revenue from more than 380 billion yuan in 2014 to 1721 billion yuan in 2018, with an average compound annual growth rate of 46.5% [2]. Many people have described China's internet industry as an unusual market that is independent from the rest of the world market, since it did not experience an early non-commercial stage but instead directly entered a rapid process of commercialization due to the involvement of the capital market [3]. Chinese internet businesses have been the subject of some controversy regarding the potential copying of technology and product development, but significant resources have been devoted to R&D. In 2017, the R&D investment of Alibaba, Tencent, Baidu, and other Chinese internet businesses totaled 106.01 billion yuan, an increase of 41.4% over the previous year. For the same period, R&D personnel reached 197,000, accounting for 19.4% of the total number of people employed in this industry [2]. Obviously, China's internet industry is an important high-tech industry with high growth and significant research and development activity.

Many internet companies are now faced with the problem of how to overcome financing constraints with the continuous expansion of the scale of the internet industry and the increase of R&D investment. Insufficient financial resources may directly require the adjustment of R&D investment and strategy, thus affecting the overall innovation realization of internet companies [4]. Because innovation requires companies to provide long-term and sufficient financial support for R&D activities, understanding how these businesses finance their investment opportunities is an important research topic in corporate financing [5]. Over the past decade, with the fundamental transformation of internet businesses from material-intensive to intangible-intensive, Chinese internet companies have obtained long-term financial support through financing innovation. However, this increase of intangible capital investment has failed to improve the overall efficiency of technological innovation for internet businesses (Fig 1). Studies on internet businesses have mainly focused on business models, competitivestrategies, internet entrepreneurship, and internet finance, but few studies have looked at the efficiency and influencing factors of financing innovation of Chinese internet companies [6].

How can this paradoxical development of Chinese internet businesses be explained? What mechanisms are applied to finance the investment opportunities of these companies? What is the overall innovation efficiency of these businesses? These are the core issues guiding the development of Chinese internet companies, however, little theoretical or practical analysis has been performed to address these questions.

To address the lack of relevant research, this study analyzed internet companies affiliated with China from 2008 to 2017. The efficiency of technological innovation was measured based on a distance function stochastic frontier analysis (SFA) model, and then this method was

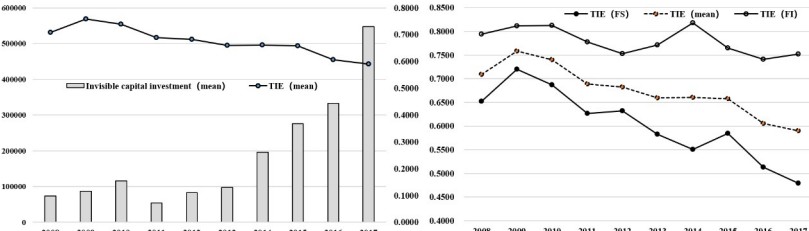

**Fig 1. Trends in average invisible capital investment and technical innovation efficiency (TIE) of listed companies in the internet industry affiliated with China from 2008 to 2017.** (left) The average numbers of invisible capital investments and TIE. (right) The average TIE for companies with financing innovation, without financing innovation, and all companies in the industry. (Drawing on the research of William Mann [7], Semyon Malamud and Francesca Zucchi [8], the increase of net debt and equity is considered financing innovation).

applied to assess the impact of financing innovation on technological innovation. In addition, we systematically examined other potential factors that affect the efficiency of technological innovation of internet businesses, and explored the effect of each factor on efficiency through SYS-GMM and panel effect models. Based on this analysis, we proposed suggestions to improve the efficiency of technological innovation.

The contributions of this work are as follows: (1) by manually sorting the practical patent data of Chinese internet companies, we examined whether financing innovation affected the efficiency of technological innovation based on the theories of financing constraints and innovation value. This work expanded the research boundary in this field, and the results provide an empirical reference to understand how financing innovation affects the efficiency of technological innovation. (2) Previous studies of internet companies mainly focused on the measurement of innovation efficiency, ignoring the potential impacts of intangible capital investment and financing innovation. We used a multiple input-output SFA model to measure the efficiency of technological innovation, and used GMM estimation to reveal the impact of financing innovation on the technological innovation efficiency of internet companies.

The remainder of this work is organized as follows. Section 2 briefly describes the two main strands of the literature that provide the basis for this work; Section 3 introduces the data, and the measurement of the efficiency of technological innovation and its analysis; Section 4 presents the empirical methodology and reports the results of the empirical analysis; Section 5 concludes and provides policy implications.

## Literature review and hypothesis development

### Technological innovation efficiency of internet companies

The rapid development of China's internet industry has aroused suspicion about "plagiarism and innovation", which is still developing rapidly [9]. Liu Jing, Fan Jingming [3] and Ana M. Fernandes et al. [10] found that Chinese internet enterprises realize technological innovation through an integrated innovation mode. Some companies have low efficiency for technological innovation, and utilize secondary innovation and creative integration to achieve breakthrough innovation. With the rapid development of the internet industry, in efforts to catch up to competitors, companies strive to quickly transplant product ideas that have already been tested by the market and then carry out integrated innovation on this basis.

In recent years, to further develop the internet industry, the government has adopted industrial policies including credit, tax, government subsidies, and the promotion of market competition to encourage industrial innovation. However, little effect has been seen for these policies. Although some companies have shown increased "quantity" of innovation, there has been little obvious change in the "quality" of innovation, and no significant improvement in the technological innovation efficiency [11, 12].

Many scholars [13–16] consider R&D efficiency and innovation performance as part of technological innovation efficiency. The logarithm value of the sum of invention and patents is generally used to represent innovation performance, but this method has not been standardized [17, 18]. It is important to standardize the important technology and service protection system of Chinese internet companies [9], rather than relying on simple summation. To study the efficiency of technological innovation, it is essential to quantify innovation input and output. To calculate the efficiency of technological innovation, some studies include intermediate output (patents, scientific monographs, and publications) and final output (sales revenue of new products) into the model [17, 19–21]. Based on previous literature, we present the following hypothesis:

Hypothesis 1: An increase of R&D investment does not necessarily improve the efficiency of technological innovation.

## Financing constraints and technological innovation efficiency

Financing and investment practices in businesses have always been the core focus of corporate finance theory, and an important practical problem of businesses. There are financing constraints due to an imperfect market, and there is a large difference between internal and external financing. Companies may not be able to pay external high financing costs, resulting in insufficient financing, which lowers investment below the optimal level and cause the company to rely too much on internal self-retaining funds [22]. This will lead to future obstacles in obtaining funds, which will affect investment in production and other aspects of enterprises, and ultimately affect total factor productivity. As China's financial development lags behind economic development [23], companies are inevitably constrained by financing, which will affect the efficiency of technological innovation [24, 25].

Financing constraints will affect the scale of assets, financing costs, and credit. The stronger the financing constraints, the more significant the effects on the corresponding financing structure [25–27]. With uncertainty and the long-term nature of technological innovation activities, high financing cost and financing structure of R&D activities will affect the ability of companies to invest in innovation projects. Previous work showed that different financing sources (mainly debt and equity) affect enterprise innovation [28, 29], but pecking order theory holds that most companies prefer internal financing rather than external financing. Sun Bo et al. [4], Chen Haiqiang, Hai Qian, Wu Kai [27], and Yuan Li, Guo Shengtie [30] proposed that the financing preference of China's listed companies does not conform to the theory of priority financing, and instead the preferred financing mode is bond financing, followed by equity financing, and finally endogenous financing. Because of the high threshold and the imperfect development of China's bond market and stock market, which typically has high fees, when listed companies need external financing, they are more inclined to pursue short-term debt financing, with little long-term debt financing [4, 11]. Therefore, companies have strong financing constraints, they may become more dependent on short-term current debt, and non-current debt financing and equity financing may increase financing costs and reduce total factor productivity [31, 32]. Thus, hypothesis H2 was proposed:

Hypothesis 2: There are significant differences in the impact of financing methods on the efficiency of technological innovation under financing constraints.

## Financing innovation and technological innovation efficiency

The biggest characteristic of internet businesses is the fundamental transformation of production function from material-intensive to intangible-intensive function. Because of the redeployment, non-exclusiveness, and low liquidity of intangible capital, these businesses often face financing constraints and difficulties [4–6, 33]. Therefore, internet companies need to obtain long-term and adequate financial support through financing innovation [8, 26]. William Mann [7] showed that patents are used as collateral to raise significant debt financing, and the pledge of patents helped innovation financing. Work from Xin Chang et al. [34] showed that debt of credit default swap (CDS) trading has a positive impact on technological innovation output (patent and patent quotation). Fama and French [35], Babenko et al. [36], McKeon [37] and Francis et al. [38] argued that large amounts of cash flow through option compensation are important sources of internal financing.

Many scholars agree that financing innovation can promote the efficiency of technological innovation of internet businesses, but some scholars propose that too much R&D material

investment and innovation failure can bring great uncertainty to businesses [4, 6, 39–41]. Based on the above analysis, we present the hypothesis H3:

Hypothesis 3: Financing innovation can improve the efficiency of the technological innovation of internet companies.

# Data and technological innovation efficiency (TIE)

## The data

In 2007, the China Securities Regulatory Commission officially required the disclosure of R&D expenditures of listed and pre-IPO companies. However, for 2007, little data is available about the R&D expenses for internet companies. Thus, we analyzed data from 2008 to 2017. Companies listed in the internet industry from United States, Shanghai, Shenzhen and Hong Kong (which are all affiliated with China) were selected as research samples. The following treatments were carried out: (1) Companies with abnormal financial performance were excluded; (2) Input or output indicators with missing values were excluded; (3) Winsorization of the 1% and 99% percentiles of the continuous variables was performed to eliminate the effects of extreme values. The resulting unbalanced panel data included 135 companies for a total of 1,306 observation points. The sample was divided into 58 financing innovation enterprises and 77 non-financing innovation enterprises (the classification method is detailed in the "Input and output variables" section). The patent data, R&D expenses, and financial indicators presented here were obtained from the State Intellectual Property Office (SIPO), the China Stock Market and Accounting Research (CSMAR) database and Wind database.

## Technological innovation efficiency of listed internet companies

There are multiple ways to measure technological innovation efficiency. The index method measures the R&D efficiency by calculating the input-output conversion ratio. The data envelopment analysis (DEA) method constructs the production frontier and measures the relative efficiency of decision-making units (DMUs). A stochastic frontier approach (SFA) estimates the frontier production function by setting a parameter model and using an econometric method. This approach can be used to assess the fitting quality of the model and provide various statistical test values. To improve the accuracy of efficiency estimation, the SFA method is widely used to measure random errors and invalid rate terms [17, 18]. SFA fully considers the interference of random factors, resulting in greatly improved measurement accuracy. However, this kind of model is constrained by a single output, and previous studies typically used only a single output or multiple models of multiple output. DEA, multi index comprehensive scoring, and single input-output SFA methods have been widely used to study internet enterprises [42, 43]. This work draws on the study of Afza T et al. [17], Lutz et al. [18], Hu et al. [44], and Tai-Hsin Huang et al. [45], and combines the distance function and the stochastic frontier method to construct a multiple input-output framework of a stochastic frontier model that can directly measure TIE.

**(1) Multiple input-output models for the technology-innovation efficiency measurement.** SFA fully considers the interference of random factors, which greatly improves the accuracy of the measurement. However, the model is constrained by a single output, and previous studies typically used only a single output or multiple models of multiple output. This paper draws on the studies of Afza T et al. [17], Lutz et al. [18], Hu et al. [44] and Tai-Hsin Huang et al. [45], and combines the distance function and the stochastic frontier method to construct a multiple input-output framework of a multi stochastic frontier model that can

directly measure TIE. The output distance function based on the Cobb-Douglas function is:

$$D_o(X, Y) = f(X, Y, \delta)e^{\lambda+v} \tag{1}$$

Where, $X$ is the input variable vector; $Y$ is the output variable vector; $\delta$ is the parameter to be estimated; $\lambda$ is the fixed effect; $v$ is the random disturbance term, $v \sim$ iid $N(0, \sigma_v^2)$ The translog distance function based on the Cobb-Douglas function is:

$$\begin{aligned}
\ln D_{oit} = \alpha_0 + \sum_{m=1}^{M}\alpha_m \ln y_{it}^m + \frac{1}{2}\sum_{m=1}^{M}\sum_{k=1}^{M}\alpha_{mk}\ln y_{it}^m \ln y_{it}^k + \sum_{j=1}^{J}\ln x_{it}^j \\
+ \frac{1}{2}\sum_{j=1}^{J}\sum_{h=1}^{J}\beta_{jh}\ln x_{it}^j \ln x_{it}^h + \sum_{m=1}^{M}\sum_{j=1}^{J}\gamma_{mj}\ln y_{it}^m \ln x_{it}^j + \lambda_i + v_{it}
\end{aligned} \tag{2}$$

Where, $D_{oit}$ is the output distance function (i.e., the unobservable dependent variable); $i$ is the DMU of the $i$th sample; $y_{it}^m$ and $y_{it}^k$ are the $t$ output variables; $x_{it}^j$ and $x_{it}^j$ are the $t$ input variables.

According the linear homogeneity of the output of the distance function and the non-negative random term ln $D_{oit}$. Eq (3) is inferred by the linear homogeneity of the distance-function output and the non-negative random term ln $D_{oit}$ to solve the problem when $D_{oit}$ is unobservable in Eq (2):

$$\begin{aligned}
\ln\left(\frac{D_{oit}}{y_{it}^M}\right) = \alpha_0 + \sum_{m=1}^{M-1}\alpha_m \ln y_{it}'^m + \frac{1}{2}\sum_{m=1}^{M}\sum_{k=1}^{M}\alpha_{mk}\ln y_{it}'^m \ln y_{it}'^k + \sum_{j=1}^{J}\beta_j \ln x_{it}^j \\
+ \frac{1}{2}\sum_{j=1}^{J}\sum_{h=1}^{J}\beta_{jh}\ln x_{it}^j \ln x_{it}^h + \sum_{m=1}^{M}\sum_{j=1}^{J}\gamma_{mj}\ln y_{it}'^m \ln x_{it}^j + \lambda_i + v_{it}
\end{aligned} \tag{3}$$

where $y_{it}'^m = y_{it}^m / y_{it}^M$, $m = 1,\ldots,M-1$, Eq (3) can be rewritten as:

$$\begin{aligned}
-\ln y_{it}^M = \alpha_0 + \sum_{m=1}^{M-1}\alpha_m \ln y_{it}'^m + \frac{1}{2}\sum_{m=1}^{M}\sum_{k=1}^{M}\alpha_{mk}\ln y_{it}'^m \ln y_{it}'^k + \sum_{j=1}^{J}\beta_j \ln x_{it}^j \\
+ \frac{1}{2}\sum_{j=1}^{J}\sum_{h=1}^{J}\beta_{jh}\ln x_{it}^j \ln x_{it}^h + \sum_{m=1}^{M}\sum_{j=1}^{J}\gamma_{mj}\ln y_{it}'^m \ln x_{it}^j + \lambda_i + v_{it} - \ln D_{oit}
\end{aligned} \tag{4}$$

Assume that the independently distributed non-negative random term $u_{it}$, truncated at the zero of $N(u, \sigma_u^2)$, and independently distributed term $v_{it}$, replaces the unobservable component $-\ln D_{oit}$. The predicted output distance of the ith DMU is $\hat{D}_{oit} = \exp(-u_{it})$, but is only part of the error term. The predicted value of the output distance function is obtained using the given conditional expectation of $\varepsilon_{it} = v_{it} + u_{it}$.

$$\hat{D}_{oit} = E[\exp(-u_{it})|\varepsilon_{it}] \tag{5}$$

Eq (4) and Eq (5) use the maximum likelihood method to estimate parameters, and then through the conditional distribution $u|\varepsilon$ of $u$, the conditional expectation $E(e^{-u_{it}}|\varepsilon_{it})$ is obtained as an estimate of the technical innovation efficiency $TIE_{it}$, which is a number between 0 and 1. The smaller the $u_{it}$ is, the larger $TIE_{it}$, indicating the higher the technological innovation efficiency.

**Table 1. Definitions of input-output variables.**

| Type | Variable | Name | Definition |
|---|---|---|---|
| **Output variable** | $Lny^1$ | Number of patents | Ln (Number of patents) |
| | $Lny^2$ | Main business income | Ln (Main business income) |
| **Input variable** | $Lnx^1$ | Invisible capital investment | Ln ((administrative expenses + selling expenses) *0.3+ R&D expenses) [6, 46, 47] |
| | $Lnx^2$ | R&D manpower investment | Ln (Number of employees) |
| | $Lnx^3$ | Fixed asset investment | Ln (Fixed assets) |

Using Eq (4), we can build a multiple input-output model of internet enterprise TIE:

$$-\ln y_{it}^2 = \alpha_0 + \alpha_1 \ln\frac{y_{it}^1}{y_{it}^2} + \frac{1}{2}\alpha_2\left(\ln\frac{y_{it}^1}{y_{it}^2}\right)^2 + \sum_{j=1}^{3}\beta_j\ln x_{it}^j + \frac{1}{2}\sum_{j=1}^{3}\sum_{h=1}^{3}\beta_{jh}\ln x_{it}^j\ln x_{it}^h$$

$$+ \frac{1}{2}\sum_{k=1}^{3}\gamma_k\ln x_{kit}\ln\frac{y_{it}^1}{y_{it}^2} + \lambda_{it} + (v_{it} - u_{it})$$

(6)

**(2) Input and output variables.** This research relied on results from the studies of Qi Sun et al. [6], Afza T et al. [17], Lutz et al. [18], and Peters and Taylor [33] to select two output variables and three input variables. Eq (5) and Eq (6) were used to calculate the efficiency and inefficiency of internet enterprise technology innovation. The selected input and output variables are listed in Table 1.

As shown in Table 2, the average number of patents and main business income is 6.501 and 10.359, respectively. The corresponding standard deviation is relatively small and the data fluctuation range is small, but there is a large difference between the minimum value (0,2.268) and the maximum value (9.045,16.811). The mean invisible capital investment and fixed-assets investment values were 9.206 and 9.102, respectively, and the corresponding standard deviations were relatively small. However, the standard deviations and the difference between minimum and maximum values for technical personnel input are quite large, indicating that this variable fluctuates within a wider range.

**(3) The estimation results of multiple input-output SFA method.** The results obtained by Eq (6) are shown in Table 3. Most of the input variables' quadratic terms and interaction terms are significant. We know from the significance of the input-output variable coefficients and the fitting coefficients of the model that the SFA method is applicable and there is a technical inefficiency term ($u_{it}$). In the stochastic frontier regression model in the form of Translog function, the Wald test passes the significance test, the fitting effect of the model is good, and Sigma_u and $\lambda$ at 1% level are significant. Thus, the SFA method can be used to estimate the efficiency of technological innovation. It is important to note that for measurement of efficiency of technological innovation, multiple input-output models and multi-input-single-output models can be used. The technological innovation efficiency and inefficiency distribution

**Table 2. Descriptive statistics of input-output variables.**

| Type | Variable | Mean | S.D. | Min | Max |
|---|---|---|---|---|---|
| **Output variable** | $Lny^1$ | 6.501 | 2.731 | 0 | 9.045 |
| | $Lny^2$ | 10.359 | 1.911 | 2.268 | 16.811 |
| **Input variable** | $Lnx^1$ | 9.206 | 1.828 | 5.477 | 16.993 |
| | $Lnx^2$ | 7.198 | 1.490 | 2.639 | 14.691 |
| | $Lnx^3$ | 9.102 | 2.228 | 2.159 | 15.540 |

Table 3. Results of the multiple input-output SFA method.

| Variable | Sample1 (Full sample) | | Sample 2 (Financing innovation enterprises) | | Sample 3 (Non-financing innovation enterprises) | |
|---|---|---|---|---|---|---|
| | Coefficients | t-value | Coefficients | t-value | Coefficients | t-value |
| Constant | -431.053 | -1.10 | -5.0114*** | -3.46 | 5.0533*** | 5.13 |
| $y' = ln(y^2/y^1)$ | 0.5774*** | 33.35 | 0.8762*** | 38.05 | 0.3976*** | 4.48 |
| $0.5*y'y'$ | 0.0517*** | 6.57 | -0.0064*** | -1.54 | 0.0390*** | 5.54 |
| $lnx^1$ | 0.4773*** | 7.99 | 0.1936*** | 3.96 | 0.4373** | 2.28 |
| $lnx^2$ | -0.7438*** | -4.46 | -0.8254*** | -7.21 | 0.1031 | 0.53 |
| $lnx^3$ | -0.5415*** | -5.39 | -0.6142*** | -6.22 | -0.4745*** | -3.87 |
| $0.5*lnx^1 lnx^1$ | -0.0680** | -2.20 | -0.0363*** | -4.01 | -0.0219 | -1.07 |
| $0.5*lnx^2 lnx^2$ | 0.1025*** | 3.72 | 0.0886*** | 2.84 | 0.0722** | 2.20 |
| $0.5*lnx^3 lnx^3$ | 0.0794*** | 6.28 | 0.0303 | 1.31 | 0.0866*** | 5.61 |
| $0.5*y'lnx^1$ | -0.0068* | -1.85 | -0.0366*** | -3.45 | 0.0013 | 0.09 |
| $0.5*y'lnx^2$ | 0.0414*** | 2.77 | 0.0821*** | 3.60 | 0.0455* | 1.75 |
| $0.5*y'lnx^3$ | -0.0436* | -1.68 | -0.0096*** | -0.36 | -0.0388*** | -3.18 |
| $lnx^1 lnx^2$ | -0.0588*** | -3.29 | -0.1096*** | -6.46 | -0.1639*** | -2.95 |
| $lnx^1 lnx^3$ | -0.0178*** | -4.31 | 0.0273** | 1.99 | -0.0298 | -1.14 |
| $lnx^2 lnx^3$ | -0.0306*** | -20.67 | 0.0278*** | 11.47 | -0.0262 | -0.84 |
| Log likelihood | Wald chi(2) = 2.25e+07 | | Wald chi(2) = 5.01e+07 | | Wald chi(2) = 1394.93 | |
| Sigma_v | 2.66e-08 | 0.02 | 0.00004 | 0.70 | 0.4352*** | 22.20 |
| Sigma_u | 16.7239** | 2.21 | 1.6022*** | 8.06 | 0.3675 | |
| $\lambda = \sigma_u/\sigma_v$ | 6.30e+08*** | 8.3e+07 | 44060.15*** | 2.2e+05 | 0.8444 | |
| N | | 1306 | | 536 | | 770 |

***, **, * indicate significance at the 1%, 5% and 10% levels, respectively; log likelihood is the chi-square value.

of multiple input-output models typically are close to a normal distribution, but those of a multi-input-single-output model typically are close to a right-sided thick distribution.

## Technological innovation efficiency (TIE) of internet companies

**(1) Overview of TIE for internet industry companies.** In the study period (2008–2017), the overall efficiency was generally more than 0.50, which is higher than the average of other industries in China (mean TIE in other industries is below 0.50), but exhibited a downward trend. As shown in Fig 2, financing innovation businesses (n = 58) had a higher TIE, followed

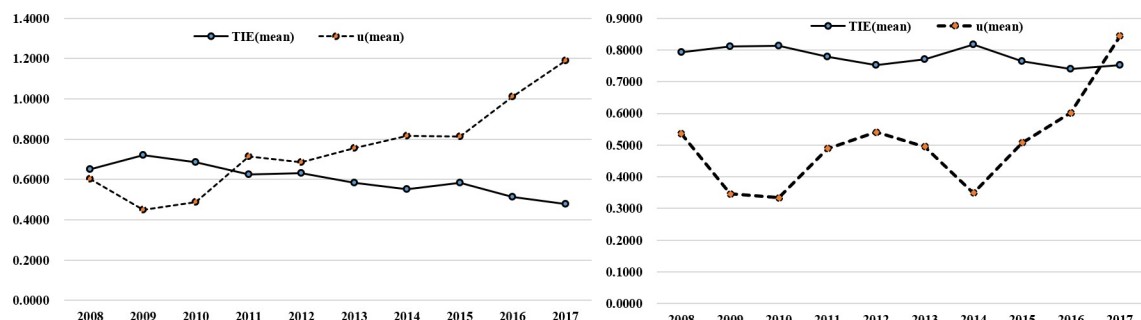

**Fig 2. Technological innovation efficiency and inefficiency trends of financing innovation enterprises and non-financing innovation enterprises during 2008–2017.** (left) The average technological innovation efficiency and inefficiency of companies with non-financing innovation. (right) The average of financing innovation enterprises.

**Table 4. The ANOVA of mean differences of the full sample.**

| Variable | Type | Sum of square | Mean square | F-value | Sig. |
|---|---|---|---|---|---|
| TIE * FI | Between groups | 10.114 | 10.114 | 134.212 | 0.000 |
| | Group internal | 98.269 | 0.075 | | |
| | Total | 108.383 | | | |
| u * FI | Between groups | 23.729 | 23.729 | 43.382 | 0.000 |
| | Group internal | 713.253 | 0.547 | | |
| | Total | 736.983 | | | |
| TIE * year | Between groups | 3.285 | 0.365 | 4.5 | 0.000 |
| | Group internal | 105.09 | 0.081 | | |
| | Total | 108.38 | | | |
| u * year | Between groups | 34.217 | 3.802 | 7.011 | 0.000 |
| | Group internal | 702.76 | 0.542 | | |
| | Total | 736.982 | | | |

by non-financing innovation businesses (n = 77). Since 2014, the technological innovation efficiency of all businesses in the sample exhibited a downward trend, and the inefficiency increased obviously.

Table 3 and Fig 2 show that the intangible capital investment ($Lnx^1$) is significantly positive, but the technological innovation efficiency (TIE) shows a downward trend, which shows that the increase of enterprise R&D investment does not necessarily improve the efficiency of technological innovation, supporting Hypothesis 1.

**(2) Mean difference analysis of technological innovation efficiency.** Table 4 shows that the F values of TIE and u at the level of financing innovation (*FI*) are 134.2 and 43.28, which are different at the level of 1% significance. The F values at the year level are 4.5 and 7.01, with differences at the 1% significance level. Table 4 and Fig 2 show that there is a significant difference in the TIE between companies that utilize financing innovation or non-financing innovation. The use of financing innovation can improve the technological innovation efficiency (TIE), so the empirical results support Hypothesis 3.

## Empirical analysis and discussion

First, descriptive statistics were made from the data for variables about enterprise financing structure, financing innovation, and control variables. Next, the correlations between variables were investigated to control for multicollinearity between variables. The SYS-GMM estimation method was then applied based on previously published work and descriptive statistical results to explore the factors affecting the financing innovation and the efficiency of technological innovation in the internet industry.

### Variables

To test the impact mechanism of enterprise financing innovation on technological innovation efficiency, factors of financing structure were selected based on work from Qi Sun et al. [6], Semyon Malamud and Francesca Zucchi [8], Afza T et al. [17], Lutz et al. [18], Yu Minggui et al. [31], and Lin Xiaoling et al. [32]. The following factors were selected: endogenous financing (*Infund*), floating debt financing (*Flodebt*), long-term borrowing financing (*Lbankg*), long-term bond financing (*Lbond*), equity financing (*Equity*), and financing innovation (*FI*). The control variables include: asset size (*Size*), return on total assets (*Roa*), management cost rate

**Table 5. Variable definitions.**

| Type | Variable | Name | Definition |
|------|----------|------|------------|
| Interpreted variable | TIE | Efficiency of Technological Innovation | Measured by Eq (6) |
| Financing structure | Infund | Endogenous financing | Endogenous financing = Net cash flow from operating activities/ Total assets |
| | Flodebt | Floating debt financing | Floating debt financing = Current liabilities / Total assets |
| | Lbankg | Long-term borrowing financing | Long-term borrowing financing = Long-term borrowing/ Total assets |
| | Lbond | Long-term bond financing | Long-term bond financing = (Non-current liabilities-Long-term borrowing)/ Total assets |
| | Equity | Equity financing | Equity financing = (End-of-year equity+ End-of-year capital reserves)/ Total assets |
| | FI | Financing innovation | Detailed in the following part "(1) Financing Innovation" |
| Control variables | Size | Asset size | Ln (Asset size) |
| | Roa | Return on total assets | Return on total assets = Net profits/ Total assets |
| | Mcost | Management cost rate | Management cost rate = Management costs/Operating income |
| | Icd | Intangible capital intensity | Ln (Invisible capital investments/Number of people) |
| | Cain | Cash interest coverage ratio | Cash interest coverage ratio = Cash flow from operating activities / Interest payments |

(*M*cost), intangible capital intensity (*Icd*) and cash interest coverage ratio (*Cain*). The specific definitions of the variables related to this study are listed in Table 5.

**(1) Financing innovation.** A characteristic of internet businesses is that intangible capital has low redeployability, non-exclusiveness, and liquidity. Based on work from Qi Sun et al. [6], William Mann [7], Semyon Malamud and Francesca Zucchi [8] and other studies, in debt financing innovation, cash interest coverage ratio that are less than 1, but increase net debt increases. Patents can be used as collateral to obtain significant debt financing. For equity financing, through equity pledge and employee stock ownership, increasing equity is considered financing innovation. To further standardize, more than 50% of the innovation frequencies of debt and equity financing of internet businesses in the observation year (2008–2017) are related to financing of innovative businesses. The frequency table of sample businesses and financing innovation is detailed in Table 6.

**(2) Intangible capital.** Another major characteristic of internet company is the fundamental transformation of production function from material-intensive to intangible-intensive. Technological innovation investment of internet businesses includes R&D expenditure; new product development expenditure; technology introduction, digestion, and absorption; staff training; purchase of domestic technology; and technological transformation. Qi Sun et al. [6], Falato et al. [46] and van Ark et al. [47] found a more than two times greater investment of intangible capital in internet businesses compared to physical capital, with a greater proportion of intangible investment such as staff training. Fig 3 shows that from 2013, intangible capital

**Table 6. Frequencies of financing innovation.**

| Whether or not to innovate in financing | Full sample | | Debt financing | | Equity financing | |
|------|------|------|------|------|------|------|
| | Observation number | Percentage | Non-financing innovative (frequency) | Financing innovative (frequency) | Non-financing innovative (frequency) | Financing innovative (frequency) |
| Non-financing innovative enterprises | 770 | 58.96 | 685 | 85 | 120 | 650 |
| Financing innovative enterprises | 536 | 41.04 | 198 | 338 | 113 | 423 |
| Total | 1306 | 100 | 883 | 423 | 233 | 973 |

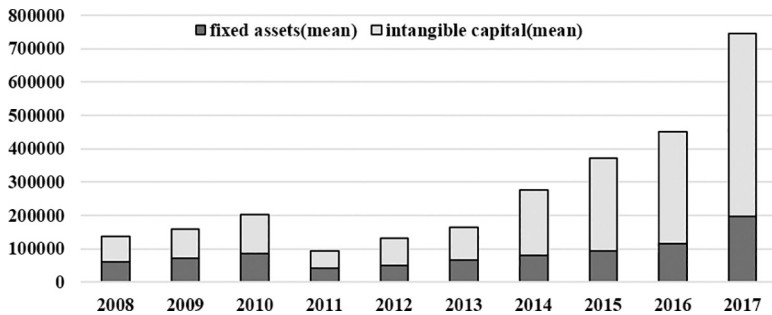

**Fig 3. The trends of fixed assets and invisible capital investment of internet enterprises during 2008–2017.**

investment was significantly larger than fixed assets, 2.47, 2.93, 2.86, and 2.78 times greater in 2014, 2015, 2016, and 2017, respectively.

## Methodology

Considering the sustainability of enterprise technological innovation efficiency and to prevent errors in the measurement model, dynamic panel system GMM estimation was used to explore the impact mechanism of the full sample and heterogeneous enterprise financing innovation on technological innovation efficiency. We established the following dynamic panel models by referring to Afza T et al. [17], Hu et al. [18], Yu Minggui et al. [31], and Lin Xiaoling et al. [32].

$$TIE_{it} = a + \beta_1 TE_{it-1} + \beta_2 Infund_{it} + \beta_3 Flodebt_{it} + \beta_4 Lbankg_{it} + \beta_5 Lbond_{it} + \beta_6 Equity_{it} + \beta_7 FI_{it} + \varepsilon_{it} \qquad \text{(Model 1)}$$

$$TIE_{it} = a + \beta_1 TE_{it-1} + \beta_2 Infund_{it} + \beta_3 Flodebt_{it} + \beta_4 Lbankg_{it} + \beta_5 Lbond_{it} + \beta_6 Equity_{it} + \beta_7 FI_{it} + \sum \beta_j x_{it} + \varepsilon \qquad \text{(Model 2)}$$

Because the profits of a company in a given year cannot be immediately invested in innovation, we delayed endogenous financing (*Infund*) for one period, and then constructed the following models:

$$TIE_{it} = a + \beta_1 TE_{it-1} + \beta_2 Infund_{it-1} + \beta_3 Flodebt_{it} + \beta_4 Lbankg_{it} + \beta_5 Lbond_{it} + \beta_6 Equity_{it} + \beta_7 FI_{it} + \varepsilon_{it} \qquad \text{(Model 3)}$$

$$TIE_{it} = a + \beta_1 TE_{it-1} + \beta_2 Infund_{it-1} + \beta_3 Flodebt_{it} + \beta_4 Lbankg_{it} + \beta_5 Lbond_{it} + \beta_6 Equity_{it} + \beta_7 FI_{it} + \sum \beta_j x_{it}$$
$$+ \varepsilon_{it} \qquad \text{(Model 4)}$$

Systematic GMM estimates the impact mechanism of enterprise financing innovation on the enterprise technological innovation efficiency, and uses the Sargan test for over-identification constraints and the Artests to test for second-order difference of perturbation terms. If Sargan_P and AR(2)_P are greater than 0.5, the systematic GMM estimation can be considered to be effective without autocorrelation; otherwise, there may be estimation errors. Given possible errors in the estimation results of the model, the corresponding fixed effect models (5)

—(8) were established to test:

$$TIE_{it} = a + \beta_1 Infund_{it} + \beta_2 Flodebt_{it} + \beta_3 Lbankg_{it} + \beta_4 Lbond_{it} + \beta_5 Equity_{it} + \beta_6 FI_{it} + u_i + \varepsilon_{it}$$
(Model 5)

$$TIE_{it} = a + \beta_1 Infund_{it} + \beta_2 Flodebt_{it} + \beta_3 Lbankg_{it} + \beta_4 Lbond_{it} + \beta_5 Equity_{it} + \beta_6 FI_{it} + \sum \beta_j x_{it} + u_i + \varepsilon_{it}$$
(Model 6)

$$TIE_{it} = a + \beta_1 Infund_{it-1} + \beta_2 Flodebt_{it} + \beta_3 Lbankg_{it} + \beta_4 Lbond_{it} + \beta_5 Equity_{it} + \beta_6 FI_{it} + u_i + \varepsilon_{it}$$
(Model 7)

$$TIE_{it} = a + \beta_1 Infund_{it-1} + \beta_2 Flodebt_{it} + \beta_3 Lbankg_{it} + \beta_4 Lbond_{it} + \beta_5 Equity_{it} + \beta_6 FI_{it} + \sum \beta_j x_{it} + u_i + \varepsilon_{it}$$
(Model 8)

Where a is a constant term and $\beta$ is a regression coefficient of each variable; $i(i = 1, 2,...,135)$ is the enterprise sample, and $t(t = 1, 2,..., 10)$ is the period; $u_i$ is the fixed effect; and $\varepsilon_{it}$ is the residual.

## Empirical results

**(1) Descriptive statistics of variables.** The results presented in Table 7 show that there is a big difference between the minimum and maximum of all variables, indicating extensive variation among different companies. The minimum values of *Infund*, *Icd*, and *Cain* are -162.02, -3.239, and -134.84, all much smaller than the maximum values. This is consistent with a big range of profitability of the different listed companies in the internet industry, with great differences in the indicators between the most profitable businesses such as Alibaba, Tencent, Baidu, and Jingdong.

**(2) Correlation coefficient analysis.** The results shown in Table 8 reveal that *TIE*, equity financing (*Equity*), and financing innovation (*FI*) are significantly positive, indicating that these factors can improve the efficiency of technological innovation, but endogenous financing

**Table 7. Descriptive statistics of variables.**

| Variable | Mean | Std. Dev. | Min | Max |
|---|---|---|---|---|
| TIE | 0.6737 | 0.2882 | 0.0304 | 0.9995 |
| u | 0.6475 | 0.7515 | 0.0005 | 6.2654 |
| Infund | -1.9049 | 9.358 | -162.02 | 7.3147 |
| Flodebt | 0.7502 | 6.1390 | 0.0003 | 169.26 |
| Lbond | 0.2212 | 0.7172 | 0 | 14.38 |
| Lbankg | 0.0887 | 0.5775 | 0.0005 | 14.204 |
| Equity | 0.1370 | 0.2276 | 0 | 5.5792 |
| FI | 0.4104 | 0.4921 | 0 | 1 |
| Mcost | 1.8643 | 41.041 | 0.0022 | 1436.058 |
| Icd | 2.0075 | 1.3411 | -3.239 | 8.4787 |
| Roa | 6.493 | 25.199 | -185.44 | 99.036 |
| Cain | 140.558 | 1220.5 | -134.84 | 19390.6 |
| Size | 11.824 | 1.7810 | 5.65617 | 18.00517 |

Source: SIPO, CSMAR Database and Wind Database.

**Table 8. Relevance of variables.**

|  | *TIE* | *Infund* | *Flodebt* | *Lbond* | *Lbankg* | *Equity* | *FI* |
|---|---|---|---|---|---|---|---|
| *TIE* | 1.0000 |  |  |  |  |  |  |
| *Infund* | -0.0538* | 1.0000 |  |  |  |  |  |
|  | (0.9740) |  |  |  |  |  |  |
| *Flodebt* | -0.0010 | 0.0127 | 1.0000 |  |  |  |  |
|  | (0.9707) | (0.6488) |  |  |  |  |  |
| *Lbond* | -0.1531*** | 0.0151 | -0.0173 | 1.0000 |  |  |  |
|  | (0.0000) | (0.5880) | (0.5321) |  |  |  |  |
| *Lbankg* | 0.0036 | 0.0035 | -0.0073 | -0.0004 | 1.0000 |  |  |
|  | (0.8957) | (0.8984) | (0.7929) | (0.9878) |  |  |  |
| *Equity* | 0.0938*** | -0.0315 | -0.0212 | 0.1177*** | -0.0206 | 1.0000 |  |
|  | (0.0007) | (0.8984) | (0.4449) | (0.0000) | (0.4561) |  |  |
| *FI* | 0.3055*** | -0.0531* | -0.0599** | -0.0788*** | -0.0223 | 0.1219*** | 1.0000 |
|  | (0.0000) | (0.057) | (0.0303) | (0.0049) | (0.4214) | (0.0000) |  |

***, **, * indicates significance at the 1%, 5% and 10%levels, respectively; the values in parentheses indicate the corresponding estimated t value.

(*Infund*) and long-term bonds financing (*Lbond*) are negative. Financing innovation (*FI*), endogenous financing (*Infund*), floating debt financing (*Flodebt*), and long-term bond financing (*Lbond*) are significantly negative, and equity financing (*Equity*) is significantly positive, indicating that financing innovation of internet businesses mainly occurs through equity incentives, employee stock ownership, and other means to support long-term, sufficient funding for R&D activities. The correlation coefficients in this model are all less than 0.4, and the test of variance expansion factor shows a maximum value 5.47, which is much less than the critical value of 10. Thus, the possibility of serious multiple collinearity among variables is excluded.

**(3) Empirical test and result analysis.** The results of SYS-GMM regression are shown in Table 9. The values were greater than 0.5 for the Sargan_P and AR(2)_P values of all models, suggesting the GMM estimation is efficient and there is no autocorrelation. The *FI* coefficients of all models are significantly positive (0.1606, 0.1847, 0.1493, and 0.1798), indicating that financing innovation can improve the efficiency of technological innovation. The *Flodebt* coefficients of all models are significantly positive (0.005, 0.0048, 0.005, and 0.0047), while long-term bond financing(*Lbond*) and equity financing(*Equity*) coefficients are significantly negative. The results show that there is a period mismatch phenomenon in the financing of technological innovation. When the financing channel is short-term debt, due to the pressure of debt repayment and the protection of shareholders' interests, companies invest in projects to improve the efficiency of technological innovation. This is consistent with previous evaluations of Chinese listed companies [31, 32], indicating that the impact of enterprise financing on technological innovation efficiency can vary under different financing constraints, supporting Hypothesis 2.

In model (2) and model (4), the *Roa* coefficients are significantly negative (-0.0012 and-0.0011). Generally, companies with poor profitability have a more urgent need to undertake technological innovation, increasing the chances of major technological breakthroughs.

Table 10 shows the results of panel regression analysis. For comparison, regression analysis of panel fixed effect and panel random effect models were also carried out. All models pass the F value and Wald tests, indicating the models are effective. According to the Hausman test

**Table 9. Results of the SYS-GMM estimation method.**

|  | Model 1 | Model 2 | Model 3 | Model 4 |
|---|---|---|---|---|
| *Constant* | 0.1455*** (4.40) | 0.3321*** (2.73) | 0.1578*** (4.52) | 0.4037*** (3.39) |
| *TIE_l* | 0.7108*** (26.64) | 0.6558*** (18.67) | 0.7074*** (26.33) | 0.6467*** (19.25) |
| *Infund* | -0.0001 (-0.88) | -0.0015 (1.64) |  |  |
| *Infund_l* |  |  | 0.0002 (1.63) | 0.0001 (0.53) |
| *Flodebt* | 0.0050*** (26.54) | 0.0048*** (11.79) | 0.0050*** (27.07) | 0.0047*** (11.64) |
| *Lbond* | -0.0264*** (-4.02) | -0.0299*** (-4.06) | -0.0260*** (-3.95) | -0.0285*** (-3.93) |
| *Lbankg* | 0.1003* (1.74) | 0.1528 (1.39) | 0.1026 (1.16) | 0.1192 (1.15) |
| *Equity* | -0.1878*** (-2.79) | -0.2211*** (-3.43) | -0.1875*** (-2.79) | -0.2579*** (-4.02) |
| *FI* | 0.1606*** (2.64) | 0.1979** (2.71) | 0.1493** (2.40) | 0.230*** (3.26) |
| *Mcost* |  | -0.0003 (-1.13) |  | -0.0003 (-1.01) |
| *Icd* |  | -0.0143 (-1.52) |  | -0.0188** (-1.97) |
| *Cain* |  | -0.0001 (1.08) |  | 0. (1.89) |
| *Size* |  | -0.1022 (-0.98) |  | -0.0155 (-1.52) |
| *Roa* |  | -0.0012*** (-3.05) |  | -0.0011*** (-2.91) |
| **Wald** | 1429.62 | 1672.26 | 1441.82 | 1701.68 |
| **Sargan_P** | 0.1911 | 0.1146 | 0.2579 | 0.2243 |
| **AR(2)** | 0.8616 | 0.5836 | 0.8611 | 0.7166 |
| **N** | 1171 | 1171 | 1171 | 1171 |

***, **, * indicates significance at the 1%, 5% and 10%levels, respectively; the values in parentheses indicate the corresponding estimated z value.

**Table 10. Results of the panel regression analysis.**

|  | Fixed effect model | | | | Random effect model | |
|---|---|---|---|---|---|---|
|  | Model 6 | | Model 8 | | Model 5 | Model 7 |
| **Variables** | FI = 1 | FI = 0 | FI = 1 | FI = 0 |  |  |
| *Constant* | 1.0086*** (9.92) | 0.9774*** (9.70) | 0.9793*** (8.04) | 1.0417*** (9.19) | 0.6004*** (30.42) | 0.5760*** (26.96) |
| *Infund* | 0.0005 (0.91) | 0.0005 (0.49) |  |  | 0.0002 (0.92) |  |
| *Infund_l* |  |  | -0.00005 (-0.19) | -0.0001 (-0.26) |  | 0.00002 (0.09) |
| *Flodebt* | -0.0595 (-0.98) | 0.0040*** (2.97) | -0.0568 (-0.86) | 0.0038*** (2.76) | 0.0037*** (3.21) | 0.0037*** (3.22) |
| *Lbond* | 0.0061 (0.18) | -0.0410*** (-3.34) | 0.0233 (0.65) | -0.0428*** (-3.37) | -0.0406*** (-4.04) | -0.0389*** (-3.76) |
| *Lbankg* | 0.1609*** (4.33) | -0.0057 (-0.45) | 0.2810*** (5.22) | -0.1717 (-0.90) | 0.0058 (0.52) | 0.2286*** (3.80) |
| *Equity* | 0.0214 (0.67) | -0.0978 (-0.44) | -0.0690 (-0.82) | -0.0762 (-0.94) | 0.0392 (1.21) | 0.0689 (1.26) |
| *FI* |  |  |  |  | 0.1752*** (5.82) | 0.1778*** (5.62) |
| *Mcost* | 0.1714*** (8.52) | 0.0003* (1.75) | 0.1686*** (8.12) | 0.0004* (1.78) |  |  |
| *Icd* | -0.0228*** (-2.61) | -0.0427*** (-4.93) | -0.0240*** (-2.61) | -0.0429*** (-4.54) |  |  |
| *Cain* | 0.0000 (1.48) | 0.0000 (0.18) | 0.0000 (1.34) | 0000 (-0.77) |  |  |
| *Size* | -0.0182** (-2.17) | -0.0241*** (-2.84) | -0.0151 (-1.57) | -0.0294*** (-3.15) |  |  |
| *Roa* | -0.0182** (-2.17) | -0.0241*** (-2.84) | -0.0151 (-1.57) | -0.0294*** (-3.15) |  |  |
| **R-sq:within** | 0.1995 | 0.0861 | 0.2224 | 0.0865 | 0.0230 | 0.0385 |
| **F-value** | 13.07 | 7.13 | 13.25 | 6.32 |  |  |
| **Wald** |  |  |  |  | 64.00 | 73.66 |
| **Hausman** | 0.0005 | 0.0001 | 0.0000 | 0.0004 | 0.9983 | 0.2192 |
| **N** | 536 | 770 | 481 | 690 | 1306 | 1171 |

***, **, * indicates significance at the 1%, 5% and 10% levels, respectively; the values in parentheses indicate the corresponding estimated t or z value.

results, models (6) and (8) use panel fixed effect models, and models (5) and (7) use panel random effect models.

For analysis of model (5) and model (7), the coefficients of financing innovation (*FI*) are significantly positive (0.1752 and 0.1778), which is consistent with SYS-GMM regression results. In all models, the current and lagging periods of endogenous financing (*Infund*) are not significantly different, indicating that the capital investment needed for technological innovation mainly depends on external financing. The results of model (6) and model (8) show that the long-term borrowing financing (*Lbankg*) coefficients of financing innovation businesses are significantly positive, the floating debt financing (*Flodebt*) coefficients of non-financing innovation businesses are significantly positive, and the long-term bond financing (*Lbond*) coefficients are significantly negative. The financing structure of financing innovation and non-financing innovation businesses can exhibit different effects on the efficiency of technological innovation. To improve the efficiency of technological innovation, internet businesses must adopt long-term borrowing financing through mortgages such as patents. The management cost rate (*Mcost*) coefficients are significantly positive, and the intangible capital intensity (*Icd*) and the total return on assets (*Roa*) coefficients are significantly negative. This is consistent with previous conclusions of reviews, indicating a need to improve the investment efficiency of internet businesses. Also, the results show that financing innovation helps to improve the efficiency of technological innovation, with significant differences in the impact of financing methods on technological innovation efficiency under financing constraints. These empirical results support Hypothesis 2 and 3.

## Robustness test of panel data model

To ensure the reliability of the empirical test results of the impact of financing innovation on technological innovation efficiency described in the previous section, the following robust tests were performed: (1) To measure technological innovation efficiency, R&D investment replaced intangible capital investment. In previous analyses, the R&D input index was widely used to measure technological innovation efficiency or innovation efficiency; (2) Firms with assets ranging from 10% to 90% were selected for analysis.

Table 11 shows that the SYS-GMM regression and fixed effect models are effective. The significance and direction of the regression coefficients of all variables are consistent with the results in Tables 9 and 10. In summary, the robustness test results did not significantly change the conclusions of empirical analysis, therefore, the conclusions of this study are basically stable.

## Conclusion and policy implications

Using the data of 135 Chinese companies in the internet industry listed at home and abroad from 2008 to 2017 (S1 Dataset), this study empirically studied the impact of financing innovation on technological innovation efficiency by using a distance function SFA model and a panel data analysis model. Application of the models provides empirical evidence to improve business efficiency and increase technological innovation. The relevant results are summarized as follows. First, there is significant variation in overall technological innovation efficiency of listed companies in the internet industry, and there is a downward trend. The technological innovation efficiency of business that use financing innovation methods is higher than those that do not. Second, the number of patents and intangible capital investment of internet businesses increase obviously every year, but there is no corresponding increase in the efficiency of technological innovation, and little intangible capital investment of non-financing innovation businesses. Thus, determining how to effectively improve the quality of patents and the

**Table 11. Results of the fixed effect model and the SYS-GMM model.**

| Variables | SYS-GMM | | Fixed effect model | |
|---|---|---|---|---|
| | | | FI = 1(Financing innovative) | FI = 0(Non-financing innovative) |
| Constant | 0.0708** (2.14) | 0.2345* (1.79) | 1.158*** (9.60) | 0.9022*** (8.22) |
| TIE_l | 0.7324*** (25.62) | 0.6648*** (20.05) | | |
| Infund | -0.0009 (-1.07) | -0.0018* (-1.84) | 0.0009 (0.50) | -0.0021* (-1.66) |
| Flodebt | 0.0724*** (9.11) | 0.0675*** (8.50) | 0.0001 (0.02) | -0.0410* (-1.96) |
| Lbond | -0.0287*** (-4.26) | -0.0316*** (-4.60) | -0.01249*** (-4.14) | -0.0244*** (-2.26) |
| Lbankg | 0.2055*** (2.66) | 0.1987** (2.29) | -0.0778 (-0.69) | -0.0043 (-0.31) |
| Equity | -0.1639*** (-2.70) | -0.2111*** (-3.49) | 0.0315 (0.82) | 0.0145 (0.25) |
| FI | 0.2256*** (3.63) | 0.2256*** (3.63) | | |
| Mcost | | -0.0002 (-0.87) | 0.0112 (1.51) | 0.0002* (1.78) |
| Icd | | -0.0164* (-1.77) | -0.0267** (-2.51) | -0.0574*** (-6.92) |
| Cain | | -0.0001 (-1.46) | 0.0001 (0.54) | 00001 (-0.75) |
| Size | | -0.0051 (-0.47) | -0.0151 (-1.57) | -0.0128 (-1.49) |
| Roa | | -0.0014*** (-3.35) | -0.0005 (-0.62) | -0.0014*** (-3.73) |
| R-q:within | | | 0.1508 | 0.1344 |
| F-value | | | 4.97 | 9.72 |
| Wald | 928.98 | 794.80 | | |
| Sargan_P | 0.2418 | 0.2837 | | |
| AR(2) | 0.1952 | 0.1232 | | |
| Hausman | | | 0.0000 | 0.0004 |
| N | 1044 | 1044 | 416 | 757 |

***, **, * indicates significance at the 1%, 5% and 10% levels, respectively; the values in parentheses indicate the corresponding estimated t or z value.

efficiency of intangible capital investment is essential to improve the overall efficiency of technological innovation for Chinese internet businesses. Third, in SYS-GMM analysis, the floating debt financing (*Flodebt*) coefficient is significantly positive and the long-term bond financing (*Lbond*) and equity financing coefficients (*Equity*) are significantly negative, suggesting a term mismatch in the investment and financing of internet businesses in China, which would force more short-term debt financing channels. Short-term debt financing investments in projects can improve the efficiency of technological innovation due to the pressure of debt repayment and the need to protect shareholders' interests. Fourth, there are different impacts of financing structure on the efficiency of technological innovation for businesses that utilize or do not utilize financing innovation. To improve the efficiency of technological innovation, internet businesses should adopt long-term financing through mortgages such as patents. In the panel regression, the coefficients of *Icd* and *Roa* were significantly negative, suggesting that the investment efficiency of internet businesses needs to be improved.

The results of the analysis suggest the following micro-suggestions: First, expansion of financing channels, through financing innovation, should reduce financing costs. Currently, China mainly relies on floating debt financing to improve the overall factor productivity of businesses. Banks are typically unwilling to lend long-term credit to innovative projects. Many internet businesses have little capital and face financing difficulties, due to the high costs of short-term credit financing [31, 32]. Using a single financing channel may lead to short-term financial risk and technological innovation efficiency of businesses. Therefore, the government should promote the development of a multi-level capital market to create a suitable financial environment for more effective development of internet businesses. Second, intangible capital investment and patent quality need to be improved. In recent years, although there have been

increased numbers of patents and intangible capital investment of internet businesses, there has been little increase in the efficiency of technological innovation. Unlimited expansion of intangible capital investment may restrict the improvement of technological innovation efficiency of businesses. It is necessary to both improve the quality of patents and increase the number of effective patents of internet businesses. Third, we should guide internet companies work to ensure productivity. If a company does not have profit support, its technological innovation activities are not sustainable. Recently, many internet businesses have focused on development without profit, leading to decreased competitiveness and technological innovation efficiency. Only through good business performance can financing be guaranteed, allowing efficient innovation investment for sustained innovation.

## Supporting information

**S1 Dataset. Data used in empirical analysis and robustness test.**
(XLS)

## Author Contributions

**Writing – original draft:** Zhefan Piao, Yueqin Lin.

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
