## [Decision Letter · Decision Letter 0]

23 Jul 2020

PONE-D-20-12599

Financing innovation and enterprises’ efficiency of technological innovation in the internet industry：Evidence from China

PLOS ONE

Dear Dr. Lin,

Thank you for submitting your manuscript to PLOS ONE. After careful consideration, we feel that it has merit but does not fully meet PLOS ONE’s publication criteria as it currently stands. Therefore, we invite you to submit a revised version of the manuscript that addresses the points raised during the review process.

The manuscript requires further revisions towards theoretical framework and quantitative approach.

We look forward to receiving your revised manuscript.

Kind regards,

Stefan Cristian Gherghina, PhD. Habil.

Academic Editor

PLOS ONE

Journal Requirements:

Reviewers' comments:

Reviewer's Responses to Questions

**Comments to the Author**

1. Is the manuscript technically sound, and do the data support the conclusions?

Reviewer #1: Yes

Reviewer #2: Yes

2. Has the statistical analysis been performed appropriately and rigorously? 

Reviewer #1: Yes

Reviewer #2: Yes

3. Have the authors made all data underlying the findings in their manuscript fully available?

Reviewer #1: Yes

Reviewer #2: Yes

4. Is the manuscript presented in an intelligible fashion and written in standard English?

Reviewer #1: Yes

Reviewer #2: Yes

5. Review Comments to the Author

Reviewer #1: The manuscript is very professionally written. Introduction is very comprehensive, background, aim, and contribution of the paper are clearly articulated. The study uses unique dataset.

Empirical model and techniques is theoretically sound. All issues are clearly identified and addresses and further robust checks are performed. Results are clearly articulated and discussed. Literature is upto date. Overall the manuscript makes important contribution to literature on impacts of financing innovation and technological progress, it is professionally written, language is appropriate.

Minor: Authors should improve the presentation of graphs and figures. Policy should be link to the finding of the study.

Reviewer #2: Firstly, the authors need to state the source of the data for China’s digital economy being the second in size in the world after the United States. Also, authors need to develop a section in the work that brings out the gap they intend to fill. That particular aspect is lacking in the work.

Secondly, there should a clear cut section that deals with highlighting the hypothesis of the study. This is necessary in order to make the objectives of the study more eligible to readers.

Thirdly, it will be interesting to have a theoretical framework to back up study objectives, as this gives a clearer picture to the methodology developed in the other section of the work.

Furthermore, there should be clear reasons as to why the samples were divided into 58 financing innovations and 77 non-financing innovation enterprises. What is the motive for such division?

It is pertinent to show clearly the side of efficiency of technological innovation in the methodology section. Authors say they divided the samples into financing and non-financing enterprises, what measures show efficiency of technological innovation? That should be clearly stated in the methodology section.

The definition of efficiency of technological innovation on table 5 is not very clear. What does it mean and how was such variable measured? (Eq1-Eq6 doesn’t show enough meaning to the variable)

Lastly, the paper could benefit from a professional editing exercise.

6. PLOS authors have the option to publish the peer review history of their article (what does this mean?). If published, this will include your full peer review and any attached files.

Reviewer #1: No

Reviewer #2: No

---

## [Author Response · Author response to Decision Letter 0]

23 Aug 2020

Replies to the reviewers’ comments:

1. The authors need to state the source of the data for China’s digital economy being the second in size in the world after the United States. Also, authors need to develop a section in the work that brings out the gap they intend to fill. That particular aspect is lacking in the work.

Response: We have added the source of data for China’s digital economy being the second in size in the world after the United States. Also, we have developed a section about the contributions of our work to state the gap we intend to fill.

2. There should a clear cut section that deals with highlighting the hypothesis of the study. This is necessary in order to make the objectives of the study more eligible to readers.

Response: We have added three hypotheses to construct a theoretical framework.

3. It will be interesting to have a theoretical framework to back up study objectives, as this gives a clearer picture to the methodology developed in the other section of the work.

Response: To answer this question, we have rewritten Chapter 2 “Literature review and hypothesis development”.

4. There should be clear reasons as to why the samples were divided into 58 financing innovations and 77 non-financing innovation enterprises. What is the motive for such division?

Response: The motive is detailed in the “Input and output variables” section.

5. It is pertinent to show clearly the side of efficiency of technological innovation in the methodology section. Authors say they divided the samples into financing and non-financing enterprises, what measures show efficiency of technological innovation? That should be clearly stated in the methodology section.

Response: The measures of technological innovation efficiency have been detailed in the methodology section. 

6. The definition of efficiency of technological innovation on table 5 is not very clear. What does it mean and how was such variable measured? (Eq1-Eq6 doesn’t show enough meaning to the variable)

Response: The efficiency of technological innovation is calculated by Eq. (6). And the variables used in Eq. (6) have been introduced in Table 1.

7. The paper could benefit from a professional editing exercise.

Response: We found a professional editing organization for help. The certificate of language editing is uploaded with manuscripts.

---

## [Decision Letter · Decision Letter 1]

3 Sep 2020

Financing innovation and enterprises’ efficiency of technological innovation in the internet industry：Evidence from China

PONE-D-20-12599R1

Dear Dr. Lin,

We’re pleased to inform you that your manuscript has been judged scientifically suitable for publication and will be formally accepted for publication once it meets all outstanding technical requirements.

Kind regards,

Stefan Cristian Gherghina, PhD. Habil.

Academic Editor

PLOS ONE

Additional Editor Comments (optional):

Reviewers' comments:

Reviewer's Responses to Questions

**Comments to the Author**

1. If the authors have adequately addressed your comments raised in a previous round of review and you feel that this manuscript is now acceptable for publication, you may indicate that here to bypass the “Comments to the Author” section, enter your conflict of interest statement in the “Confidential to Editor” section, and submit your "Accept" recommendation.

Reviewer #1: All comments have been addressed

Reviewer #2: All comments have been addressed

2. Is the manuscript technically sound, and do the data support the conclusions?

Reviewer #1: Yes

Reviewer #2: Yes

3. Has the statistical analysis been performed appropriately and rigorously? 

Reviewer #1: Yes

Reviewer #2: Yes

4. Have the authors made all data underlying the findings in their manuscript fully available?

Reviewer #1: Yes

Reviewer #2: Yes

5. Is the manuscript presented in an intelligible fashion and written in standard English?

Reviewer #1: Yes

Reviewer #2: Yes

6. Review Comments to the Author

Reviewer #1: I am satisfied with all the revision and do not have any further comments. I would like to congratulate the authors.

Reviewer #2: I the paper has been significantly improved upon. Hence I have no further worries about the paper. I will therefore recommend that it is accepted for publication.

7. PLOS authors have the option to publish the peer review history of their article (what does this mean?). If published, this will include your full peer review and any attached files.

Reviewer #1: **Yes: **Baljeet Singh

Reviewer #2: No

---

## [Editor Report · Acceptance letter]

7 Sep 2020

PONE-D-20-12599R1 

Financing innovation and enterprises’ efficiency of technological innovation in the internet industry: Evidence from China 

Dear Dr. Lin:

I'm pleased to inform you that your manuscript has been deemed suitable for publication in PLOS ONE. Congratulations! Your manuscript is now with our production department. 

Kind regards, 

on behalf of

Dr. Stefan Cristian Gherghina 

Academic Editor

PLOS ONE